# EXPLOITING ACTION DISTANCES FOR REWARD LEARNING FROM HUMAN PREFERENCES

## ABSTRACT

Preference-based Reinforcement Learning (PbRL) with binary preference feedbacks over trajectory pairs has proved to be quite effective in learning complex preferences of a human in the loop in domains with high dimensional state spaces and action spaces. While the human preference is primarily inferred from the feedback provided, we propose that the policy being learned (jointly with the reward model) during training can provide valuable learning signal about the structure of the state space that can be leveraged by the reward learning process. We introduce an action distance measure based on the policy and use it as an auxiliary prediction task for reward learning to influence its embedding space. This measure not only provides insight into the transition dynamics of the environment but also informs about the reachability of states and the overall state space structure. We evaluate the performance and sample efficiency of our approach using a combination of six tasks in Meta-World domains with simulated oracles. We also conduct human in the loop evaluation on three tasks to confirm our findings from oracular experiments. We demonstrate that the proposed simple auxiliary task for constraining reward model's embedding space can provide strong empirical improvements to sample efficiency and accelerate policy learning.

## 1 INTRODUCTION

Preference-based Reinforcement learning (PbRL) is a promising paradigm for training agents to learn from human preferences Leike et al. (2018); Akrour et al. (2011); Ibarz et al. (2018b); Bakker et al. (2022); Köster et al. (2020). While human feedback can be obtained and incorporated in several ways, the primary objective of PbRL is to distill information from binary preference feedbacks on queried agent behavior trace pairs into the reward function Wilson et al. (2012); Christiano et al. (2017). Recent advances in PbRL have led to algorithms that are capable of successfully learning human preferences on simpler discrete tasks Verma & Metcalf (2022); Soni et al. (2022) to more complex continuous control tasks Lee et al. (2021a); Park et al. (2022). A key challenge in PbRL is to reduce human feedback sample-complexity. Typically, prior works have investigated research directions like improving the query sampling strategy Lee et al. (2021a), performing state or trajectory augmentation in the queries Park et al. (2022); Guan et al. (2021), unsupervised policy pre-training Lee et al. (2021a), learning world models as reward priors Verma & Metcalf (2022). However, exploiting the agent policy being learned along with the reward model has not been explored before specifically in PbRL.

Our main intuition is that the learned policy function could very well be utilized to improve the reward learning process since it can inform about distance between states or towards a preferred state as indicated by the human preference (or more generally the state space organization and reachability among states). In the existing line of research Park et al. (2022); Lee et al. (2021a); Christiano et al. (2017), the reward learning process can only distinguish between the states based on the given state representation as temporal features are not provided to the agent. While the existing state features maybe enough to eventually (given enough feedback) distinguish between goodness of states we show that having access to a more suitable feature that informs about distance between states can lead to significant improvements to feedback efficiency and policy acceleration.

A policy trained on the reward model (which itself is non-stationary and is updated iteratively), would encapsulate both of our sought-after attributes. For example, a bank of sampled trajectories using

the said policy contains information about which states could be reached from which other states, i.e., **reachability**, which actions would lead the agent to be in a certain part of the Markov Decision Process (MDP), i.e., **environment dynamics**. And, finally, since the policy was trained to maximize returns predicted by the reward model, a good policy starting from any state would sample trajectories where successor states are also according to human preference i.e., **improved quality of states in query buffer**. We intend to extract these key pieces of information available to the agent via policy learning to improve the reward learning in PbRL as shown in Figure 1.

We propose a self-supervised method for learning the distance (action distance) between states using a jointly trained policy function, which captures weak learning signals of the intermediate policy function trained on the reward model. To improve PbRL using this action distance measure, we propose that the reward model being learned (particularly an embedding space in the reward model) must be predictive of this action distance. We operationalize this by treating the action distance prediction objective as an additional auxiliary task for the reward model, thus forcing the embedding space to preserve the action distance between any pair of states. Our experiments on six continuous control robotic manipulation tasks (Meta-World), commonly used in recent PbRL works Park et al. (2022); Lee et al. (2021a), show that the use of action distance based auxiliary task in the reward learning process is an effective means of boosting the agent's performance when learning weak learning feedback such as binary evalation in PbRL.

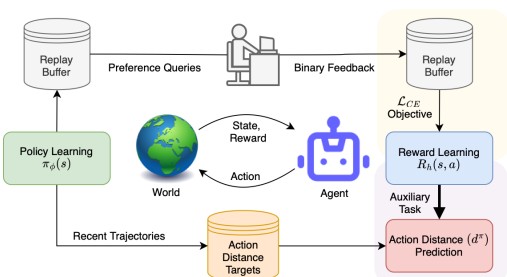

Figure 1: Overview of the proposed approach. Typical to PbRL setups, our agent is acting in the world & saving state, action, and reward data into its buffer. Every few episodes, the human in the loop (HiL) is queried for their preference over the agent's behavior trajectory. The aim of the agent is to learn a reward model (reward learning) along with obtaining a good policy (policy learning). We are proposing a method for preference-based reinforcement learning (PbRL) that uses an action distance measure based on the policy being learned as an auxiliary prediction task for reward learning.

We highlight the main contributions of the work as follows :

1. This is the first work leveraging valuable learning signal from the joint policy being learned to improve reward learning in the context of Preference Learning from binary feedback.
2. We propose an action distance based auxiliary task for the reward model that can be easily incorporated into any PbRL algorithm.
3. We benchmark our work against state-of-the-art PbRL algorithms, as well as adapted PbRL algorithms that share certain characteristics of action distance based auxiliary task.

The paper is structured as follows: we give an overview of the existing literature in Preference-based Reinforcement Learning and distance measures-based representation learning in Section 2, discuss the preliminaries to this work in Section 3, followed by our proposed methodology in Section 4. We show a comprehensive set of empirical evaluation and results in Section 5 and conclude this work in Section 6. Appendix, Ethics statement and Implemented code can be found in the supplementary.

## 2 RELATED WORK

**Preference-based Reinforcement Learning.** There are several works in the RL literature Liu et al. (2023); Bewley & Lecue (2021); Zhang & Kashima (2023); Liu & Chen (2022); Wirth et al. (2017) that focus on acquiring ratings or feedback from the human-in-the-loop Knox & Stone (2009); Christiano et al. (2017); Ibarz et al. (2018a); Stiennon et al. (2020). Knox & Stone (2009) was one of the foremost works to incorporate human-in-the-loop binary feedback to aid the agent's learning for solving the problem of sparse environment reward. However, the framework presented in Knox & Stone (2009) and further extensions of it were restricted to querying the user over state preferences. Christiano et al. (2017) proposed a deep RL framework that queried the human user for trajectory preferences instead, by asking the user to choose the preferred trajectory over the other based on the Bradley-Terry model Bradley & Terry (1952), which has further been incorporated in several works that followed Lee et al. (2021a); Park et al. (2022); Guan et al. (2022); Liu et al. (2023); Liang et al. (2022). While Lee et al. (2021a) proposed to use unsupervised pre-training to query diverse behaviors to the teacher, Park et al. (2022) utilized data augmentation techniques to learn the reward model

combined with a semi-supervised learning approach to utilize the unlabeled trajectories. Both of these works jointly learn the reward and the policy model for the agent. Ibarz et al. (2018a) combined using trajectory preferences with expert demonstrations. Finally, works in IRL Ho & Ermon (2016); Ghasemipour et al. (2020); Ni et al. (2021); Wang et al. (2019); Liu et al. (2020) attempts to "match" the agents state-action distribution with the expert's, however PbRL does not assume access to expert policy, just their feedback on queried trajectory pairs. Therefore, in PbRL the human in the loop does not control the trajectory to provide their evaluation on, nor do they provide action/policy advice. We extend the PbRL literature two key ways. First we propose cheap-to-compute distance measure (action distance) obtained through the joint policy that is a useful learning signal for reward learning. Second, we provide a simple and extendable approach to incorporating this informative feature into reward learning (auxiliary prediction task from the embedding space).

**Representation learning using distance measures.** Several works within and outside the field of reinforcement learning have focused on learning distance-guided representations that allow for learning structured representations Florensa et al. (2019; 2018); Pong et al. (2019); Nair et al. (2018) which prove to be useful for the downstream tasks. For example, Li et al. (2020) uses a distance-guided measure for representation learning for Graph Neural Networks to capture the distance between nodes for a richer expressive power. Similarly in the field of computer vision tasks, such representation learning techniques have proven to be useful Shen et al. (2018); Roh et al. (2021). Kemertas & Aumentado-Armstrong (2021) is another technique that uses bisimulation metrics (dependent on transition and reward function) improve the policy learning. Using bisimulation metrics with PbRL can be challenging as : first, the reward learning is the intended objective of PbRL and it is non-trivial to use a low-confidence predicted reward value to improve the reward function itself. Moreover, in contrast to improving the embedding space of the policy function through rewards, here the goal is to improve the embedding space of the reward function. In section 5 we show that an adapted bisimulation metric based baseline is not effective in PbRL.

It is known that guidance or heuristics informing about the goal, or distance-to-goal can help with both known Hart et al. (1968); Browne et al. (2012); Zhong et al. (2013); Hoeller et al. (2020); Bejjani et al. (2018) and unknown MDPs Cheng et al. (2021); Wagener et al. (2021); Guan et al. (2021); García & Fernández (2015). While several prior works have utilized the notion of action distances or commute times, closest to our work would be research in goal-conditioned RL Hartikainen et al. (2019) and Venkattaramanujam et al. (2019). This first work proposes to use action distances for skill learning while the second uses it for learning goal conditioned policies. Although Hartikainen et al. (2019) does discuss ways of incorporating human preferences to obtain the reward model there are several key distinctions. First, both Hartikainen et al. (2019) and Venkattaramanujam et al. (2019) require goal proposals, that is they explicitly set states as goals either by using their learned distance measure or by asking humans to label possible goals. In contrast our method does not require explicit goals. Second, both works learn an explicit distance function to approximate action distance. Our proposed solution instead shows that we can utilize the reward model's embedding space to compute the action distance. Third, Hartikainen et al. (2019) uses the computed action distance directly as the reward value which makes it incompatible to use with other PbRL techniques like reward priors Verma & Metcalf (2022) etc. Finally, while Venkattaramanujam et al. (2019) is interested in learning goal-conditioned policies, in this work, we explore the use of action distances on trajectory based binary preferences.

## 3 PRELIMINARIES

### 3.1 PREFERENCE-BASED REINFORCEMENT LEARNING

Reinforcement learning allows for agents interacting in an environment $\mathcal{E}$ where at each discrete time-step $t$, the agent receives an observation $o_t$ from the environment and chooses an action $a_t$ based on its policy $\pi$. As in conventional RL frameworks, we assume that the underlying system is a Markov Decision Process, i.e. the tuple $< \mathcal{S}, \mathcal{T}, \mathcal{A}, \tilde{\mathcal{R}}_h, \gamma >$ describing the state space $\mathcal{S}$, agent's action space $\mathcal{A}$, the underlying environment transition dynamics $\mathcal{T}$, the discount factor $\gamma$ where the agent's goal is to maximize the return $\sum_{k=0}^{\infty} \gamma^k \tilde{\mathcal{R}}_h(s_{t+k}, a_{t+k})$ computed over the reward system $\mathcal{R}_h$ in concern. In the PbRL setup that we are interested in, the goal of the agent is twofold: (1) to infer the human's underlying reward model $\tilde{\mathcal{R}}_h$ via binary feedback over trajectory pairs, and (2) to further use the learned reward model $R_h$ to compute a policy $\pi_\phi$ parameterized by $\phi$ to maximize discounted cumulative return over $R_h$.

We utilize the formulation presented in Wilson et al. (2012) for the Preference-based Reinforcement Learning problem where the agent queries the human in the loop (HiL) with a trajectory pair $(\tau_0, \tau_1)$, where, $\tau_i = \{(s_k, a_k), (s_{k+1}, a_{k+1} \cdots (s_{k+H}, a_{k+H}))\}$, for a binary feedback $y \in \{0, 1\}$ indicating their preferred trajectory. Such feedbacks along with the queried trajectories are stored in a dataset $D_\tau$ as tuples $(\tau_0, \tau_1, y)$. Following the Bradley Terry model Bradley & Terry (1952) to compute the probability of one trajectory being preferred over another, recent line of works like Lee et al. (2021a); Christiano et al. (2017) approximates the human reward function as $R_h$, parameterized by, say, $\psi$, by solving a supervised learning problem where the returns computed over the learned reward function are higher for trajectories that were preferred by the HiL than the returns computed on the non-preferred trajectory. This is done by minimizing the cross-entropy between the predictions and ground truth human labels as follows:

$$\mathcal{L}_{CE} = - \mathbb{E}_{(\tau_0, \tau_1, y) \sim \mathcal{D}} [y(0) \log P_\psi[\tau_0 \succ \tau_1] + y(1) \log P_\psi[\tau_1 \succ \tau_0]] \tag{1}$$

where probabilities $P_\psi$ are computed using the approximated reward function $R_h$ as:

$$P_\psi[\tau_0 \succ \tau_1] = \frac{\exp(\sum_t R_h(s_t^0, a_t^0))}{\sum_{i \in \{0,1\}} \exp(\sum_t R_h(s_t^i, a_t^i))} \tag{2}$$

## 3.2 Multi-dimensional Scaling

Multi-dimensional Scaling (MDS) Borg & Groenen (2005); Young & Hamer (2013) is a form of non-linear dimensional reduction where dissimilarities between pairs of the data in the original space are mapped to distances and are preserved in the low-dimension space. While MDS has typically been used to visualize similarity or dissimilarity between a set of objects in a low-dimensional space, it has also been used to construct embedding space for a set of objects by finding a set of coordinates in the low-dimensional space that minimize the difference between the distances between the objects in the original high-dimensional space and the distances between the objects in the low-dimensional space.

Classical MDS (or Toegerson-Gower Scaling, or Principal Coordinates Analysis, or cMDS) assumed that the dissimilarities in the original dimensionality are in the Euclidean space and therefore, algorithms for classical MDS preserve the input dissimilarities when these dissimilarities are Euclidean distances. Say the dissimilarity for objects $i, j$ is given by $\delta_{i,j}$, and the embedding space is given by $E$. Then, cMDS would reduce a loss (also called stress), as follows:

$$\sigma_{cMDS}(E) = \sum_{i<j} w_{ij} (d_{ij}(E) - \delta_{ij})^2 \tag{3}$$

In this work we use a generalization of classical MDS, called Metric MDS (mMDS) where in the stress Equation 3, the distance metric and dissimilarity measures are replaced by $f(x)$ to give the following stress objective:

$$\sigma_{mMDS}(E) = \sum_{i<j} w_{ij} (f(d_{ij}(E)) - f(\delta_{ij}))^2 \tag{4}$$

We note that, as pointed out in Venkattaramanujam et al. (2019) mMDS does not admit an analytical solution, instead it is solved iteratively and convergence to a global minimum is not guaranteed.

## 4 Methodology

In this section, we present our main contribution and discuss the key reasons why action distance measure is helpful for both reward learning and policy learning. Our main idea is to make the reward model being learned aware of the state space structure (i.e. reachability and environment dynamics) and turn improve the query trajectory buffer to contain desirable trajectories. We do so by making the reward model solve the auxiliary task of predicting action distances between two states. We will first ground the action distance measure and propose the methodology for incorporating it with any existing PbRL framework that learns the reward model via function approximation and any underlying RL algorithm (both online and offline).

## 4.1 ACTION DISTANCES

**Definition 4.1.** Average passage distance $m^\pi(s_j, a_j | s_i, a_i)$ is given by the expected number of actions required to go from $s_i$ to $s_j$ by taking $a_i$ as the first action and choosing $a_j$ as the final action upon reaching $s_j$.

$$m^\pi(s_j, a_j | s_i, a_i) = \mathop{\mathbb{E}}_{\tau \sim \pi} \left| \tau_{[(s_i, a_i) \cdots (s_j, a_j)]} \right| \tag{5}$$

**Definition 4.2.** Action distance, or commute distance, $d^\pi$ between two states $(s_i, s_j)$ and an initial action $a_i$, and final action $a_j$ under some policy $\pi_\phi(s)$ and transition dynamics $\mathcal{T}(s, a, s')$ is given by the expected number of action steps taken to reach a state $s_j$ from $s_i$ with the first executed action as $a_i$ and final chosen action $a_j$.

$$d^\pi(s_i, a_i, s_j, a_j) = \frac{1}{2}(m^\pi(s_j, a_j | s_i, a_i) + m^\pi(s_i, a_i | s_j, a_j)) \tag{6}$$

For shorthand, in the context of distances between states, we abuse the notations for states $s_i$ to mean $(s_i, a_i)$ tuple, since we always compute distances in the embedding space for which the input was state-action tuple. Action distances (commute distance) have been largely used in developing theory for time-reversible Markov chains Aldous & Fill (1995). A variant of it was proposed in Fouss et al. (2005) and more recently Venkattaramanujam et al. (2019) used action distances to generate goals for learning goal-conditioned policies faster. Action distances have been used in variety of applications, but Fouss et al. (2005) first showed that an embedding space that preserves commute distance must exist, however Venkattaramanujam et al. (2019) admits obtaining action distance based ground truth targets can be challenging.

While incorporating action distance measure directly into the reward learning process by making the reward model $\hat{r}_t$ target to be a scaled linear combination of the existing reward model $\hat{r}_{t-1}$ and the action distance measure maybe possible, it becomes extremely non-trivial and challenging for reasons like the scale of the distance measure and the predicted reward would have to be matched. A simpler approach would be to incorporate the action distance measure via mMDS by enforcing the embedding space of the reward model to be predictive of the action distance measure as an auxiliary task. Not only such a method would not suffer from noise due to explicit goal-proposals Hartikainen et al. (2019), but it also allows us to use it with other PbRL algorithms.

## 4.2 ACTION DISTANCES CAN IMPROVE REWARD LEARNING

The reward learning and the policy learning happen in an iterative manner in our PbRL framework. While the key information regarding the human preference is contained in the binary feedback preference labels given by the human-in-the-loop, we posit that the policy trained over these learned rewards could still provide weak learning signal about the human preference. The policy function ($\pi_\phi$) being trained takes into account the environment dynamics as well as the reward model ($R_h$) and can hint at the set of states which are desirable under the reward model at the time. An adaptation of Venkattaramanujam et al. (2019) would be to sample these potential future desirable states based on the action distance measure and try to improve reward model towards these sampled states, however this requires strong assumptions like a simulator that can be reset and started from any given state. Moreover, such a strategy may be extremely hard to stabilize the learning as the set of desirable states becomes a non-stationary target. Another direction could be to perform an Inverse RL or IRL over Ng et al. (2000); Arora & Doshi (2021); Ab Azar et al. (2020) step the learned policy to get another possible reward function, say, ($r_{IRL}$) and combine $R_h$ and $r_{IRL}$. But such a method could be very expensive (as PbRL itself is a form of IRL with binary feedbacks over trajectory pairs) and the obtained $r_{IRL}$ could be extremely noisy and may not offer any generalization to existing $R_h$. Another approach could be to learn the full world model Hafner et al. (2019) and use the reward model to desirable trajectories. However, learning the world model may again be very expensive. An approximation to this approach could be to learn a forward dynamics model where an additional objective of the reward model is to predict the next state observation (given the current state and action). While this approach is more feasible than explicitly learning the world model and that such an auxiliary task has been used in the context of improving the policy learning Nguyen et al. (2021); Zhang et al. (2018), our results demonstrate that this auxiliary task is a research challenge in itself and does not offer any performance improvements.

### 4.3 ACCELERATING POLICY LEARNING VIA ACTION DISTANCE

A special case of preference learning can be when the trajectory preference is such that there exists a goal state (absorbing state). In such a scenario we provide intuitions that explains policy acceleration as seen in our results (See 5).

**Definition 4.3.** A Markov Decision Process $\{\mathcal{S}, \mathcal{T}, \mathcal{A}, \mathcal{R}, \gamma\}$ is called strong-reversible if for any action $a_{ij}$ that allows the agent to transition from state $s_i$ to $s_j$ in a single step, there exists an action $a_{ji}$ that allows the agent to transition from $s_j$ to $s_i$ in one-step.

**Proposition 4.4.** *For any MDP, $\{\mathcal{S}, \mathcal{T}, \mathcal{A}, \mathcal{R}, \gamma\}$, with $\forall s \sim \mathcal{S}, R(s) \leq -1, \gamma = 1$ and an absorbing state $s_g$, the average passage distance $m^\pi(s_g|s_i)$ from any state $s_i$ to the goal state under a stochastic policy $\pi$ is a pessimistic heuristic.*

**Proposition 4.5.** *For a strong-reversible MDP, $\{\mathcal{S}, \mathcal{T}, \mathcal{A}, \mathcal{R}, \gamma\}, \gamma = 1, \forall s \sim \mathcal{S}, R(s) \leq -1$ and an absorbing state $s_g$, the action distance $d^\pi(s_i, s_g)$ between any state $s_i$ and the goal state under a stochastic policy $\pi$ is a pessimistic heuristic.*

As defined by Cheng et al. (2021), a pessimistic heuristic is one that overestimates the cost to goal (or underestimates the reward). Proving Proposition 4.4 is straightforward, since the average passage distance, by definition, gives an estimate of the number of actions required to reach the goal, which is an overestimate of minimum cost-to-go from that state to the goal.

Proof for Proposition 4.5: To prove, we must show that the commute distance from any state $s_i$ to any state $s_j$ is an overestimate of the shortest distance from $s_i$ to $s_j$. Since it is a strong-reversible MDP, the shortest distance from $s_i$ to $s_j$, say $d^*$, is the same as shortest distance from $s_j$ to $s_i$. From proposition 4.4, $m^\pi(s_i|s_j) \geq d^*$ and $m^\pi(s_j|s_i) \geq d^*$. Hence, $d^\pi(s_i, s_j) = \frac{1}{2}(m^\pi(s_i|s_j) + m^\pi(s_j|s_i)) \geq d^*$, is a pessimistic heuristic.

Further, Cheng et al. (2021) proved that pessimistic heuristics used with reward function are desirable to accelerate policy learning. This shows that for strong-reversible MDPs with absorbing states and at-least unit action costs, an action distance based heuristic is desirable for accelerating policy learning on the reward model. While we do not directly use the action distance for shaping the rewards from the reward model, the auxiliary task that requires the embedding space to preserve action distance does accelerate policy learning as confirmed empirically in Section 5.

### 4.4 UTILIZING ACTION DISTANCES FOR REWARD LEARNING

The central idea is to create an auxiliary objective for the reward learning task where the reward model $R_h$ is also predictive of the action distance between any two states. Under PbRL, the main reward learning objective is given by Equation 1. As shown by Fouss et al. (2005), we know that an embedding space where the distance between the points (i.e. the embedding of state, action pair) is proportional to the action distance exists. Hence, we resort to performing a metric Multi-dimensional scaling (see Section 3.2) using our action distance measure, i.e. ensure that the embedding space in the reward model $\hat{r}$ also reflects action distances.

Therefore, we ensure that the embedding space of the reward model, $R_e(s)$ of the state $s$, say the penultimate layer if $R_h$ is a neural network, by ensuring that the Euclidean distance between the embedding of two states $s_i$ and $s_j$ reflects the action distance $d^\pi(s_i, s_j)$. This can be achieved by minimizing the Mean Squared Error (MSE) between the computed distance in the embedding space and the action distance as follows:

$$\mathcal{L}^{ad} = \mathop{\mathbb{E}}_{\substack{s_i, a_i \\ s_j, a_j \\ d_y \sim D_{ad}}} (||R_e(s_i, a_i) - R_e(s_j, a_j)||^2 - d_y)^2 \tag{7}$$

where, $s_i, a_i, s_j, a_j$ are state action pairs in the dataset $D_{ad} = ((s_i, a_i), (s_j, a_j), d_y)$ which consists of the computed ground truth action distances between them as $d_y$.

#### 4.4.1 COLLECTING DATA FOR ACTION DISTANCE LOSS

**Proposition 4.6.** *For a stochastic policy $\pi$ that induces a stationary distribution, under a balanced sampling from the dataset obtained by from Algorithm 2, a perfect function approximator can estimate the action distance between the states $s_i$ and $s_j$.*

We leverage the trajectory bank $D_\tau$ (agent's replay buffer) to create the dataset with action distance targets, $D_{ad}$ (See Appendix 2). The key idea is that since the action distance ground truth that we want is an expectation over number of actions taken to reach $s_j$ from $s_i$, we can approximate this action distance by sampling a state $s_i, s_j \in \tau$ where $j > i, \tau \in D_\tau$ and use the number of action steps taken in the trajectory from $s_i$ to $s_j$ as the ground truth distance $d_y = |j - i|$ as described in Algorithm 2. An important note is that the distances $d_y$ in the dataset $D_{ad}$ should be from the agent's current policy $\pi_\phi$. For off-policy RL algorithms where the replay buffer, $D_\tau$, would contain trajectories sampled from a stale policy, we emulate the required behavior of the dataset $D_{ad}$ by ensuring that only the last $k$ trajectories added to the dataset $D_\tau$ are used to compute $D_{ad}$. Practically, instead of picking n samples uniformly within a trajectory, we found it better to generate all combinations of $(i, j)$ s.t. $i < j$ to populate the dataset. Refer to B.1 for the proof of 4.6.

### 4.4.2 Action Distance based PbRL Objective

We perform a linear combination of the proposed action distance based loss function $\mathcal{L}^{ad}$ the cross entropy loss $L_{CE}$, as in equation 1, (see Section 3) to get our novel reward learning objective:

$$\mathcal{L}^{reward} = \lambda_{CE}\mathcal{L}^{CE}(D_h) + \lambda_{ad}\mathcal{L}^{ad}(D_{ad}) \tag{8}$$

where $\mathcal{L}^{CE}$ is computed over $D_h$ containing the queried trajectory pairs with human binary feedbacks (mean over the samples), and $\mathcal{L}^{ad}$ is computed over the dataset of state pairs with action distance targets $D_{ad}$ created from the $k$ most recent trajectories added to $D_\tau$.

## 5 Empirical Evaluation

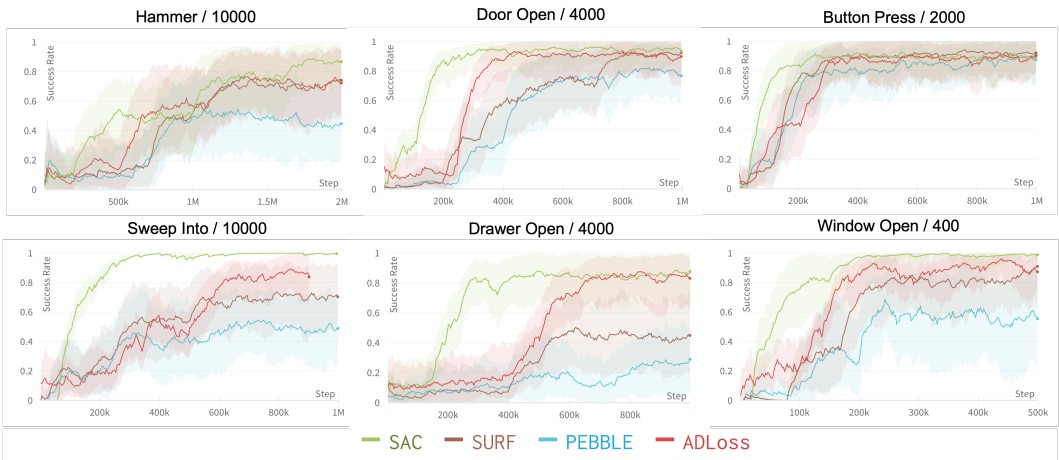

Figure 2: Learning curves on robotic manipulation tasks (given as "name / number of feedbacks") as measured on the success rate comparing ADLoss with PEBBLE, SURF and SAC.

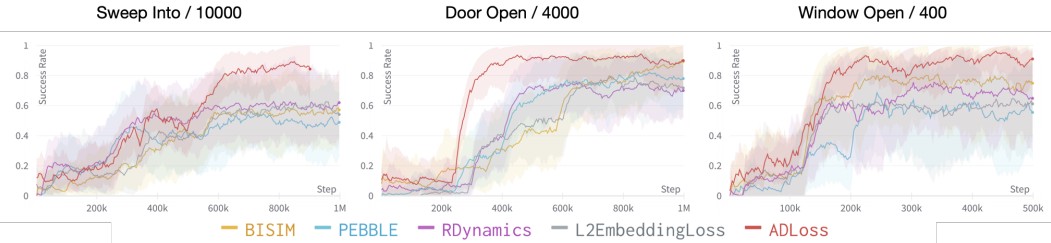

Figure 3: Learning curves on robotic manipulation tasks (given as "name / number of feedbacks") as measured on the success rate comparing ADLoss with adapted PbRL baselines Rdynamics and L2EmbeddingLoss.

We design our experiments to investigate the following:

1. How do Action Distances improve the feedback sample efficiency compared to state of the art PbRL baselines? What is it's impact on policy learning?

Figure 4: Ablation study on Button-Press (2000 feedbacks). (left) Effect of query segment length on the ADLoss agent performance for lengths $\{30, 50, 80\}$. (center) Effect of varying $\lambda_{ad}$ in equation 8 in the set $\{3, 10, 20, 40\}$. (right) Additional experiment on Sweep-Into (10000 feedbacks) to evaluate how does combining action distances with other SSL approaches may impact PbRL performance.

2. How do Action Distances fare against PbRL baselines adapted to be aware of environment dynamics, or preserve the Euclidean distance between states?

3. Impact of factors like query length and weight of the ADLoss on agent's performance?

4. Does combining ADLoss with Semi-Supervised approaches affect the overall performance?

5. Beyond simulated oracles, what is the effectiveness of ADLoss with human in the loop?

To validate our proposed method, we conduct our experiments on six domains of Meta-World, namely, Hammer, Door-Open, Drawer-Open, Window-Open, Button-Press, Sweep-Into Yu et al. (2020). The main benefits of using these domains are, (a) several existing PbRL methods have used Meta-World as a benchmark which allows for more consistent comparison with PbRL baselines (b) engineering reward functions for robotic manipulation is exceptionally hard and motivates the use of a PbRL technique, and finally, (c) as mentioned in Section 1, Meta-World. Following B-Pref Lee et al. (2021b), we consider a scripted human in the loop (HiL) who provides a binary feedback label of their preference during the agent's training. Since the oracle uses the underlying task reward (henceforth, true reward model) to generate binary feedbacks, it allows us to evaluate the learned policy on the true reward. Finally, since the proposed method can be used with any existing PbRL technique, we use the state-of-the-art approach PEBBLE Lee et al. (2021a) in our experiments as the backbone algorithm and call the combination of our reward learning objective in Equation 8 and PEBBLE as ADLoss (see Appendix C). The result plots show the mean (solid line) and standard deviation (shaded region) over five random seeds. Refer to Appendix for more details on domains D.6, experimental setup D and PEBBLE algorithm C.

## 5.1 PREFERENCE-BASED RL BASELINES

**State-of-the-art baselines:** Since PEBBLE is the backbone algorithm used in the experiments for showing the benefits of action distance measure, we use PEBBLE as baseline. We also use the state-of-the-art PbRL algorithm SURF as our baseline. We use the version of SURF without temporal data augmentation, which helps in comparing the effects of semi-supervised learning by additional trajectory samples (SURF) and semi-supervised learning of action distances (ours). Finally, we use SAC trained on underlying oracle rewards as a loose upper-bound of policy performance. Figure 2 shows the performance of action distance auxiliary task (red) to be substantially and consistently better than baselines PEBBLE and SURF. Interestingly, for several domains (Hammer, Door-Open, Button-Press, Drawer-Open, Window-Open), our algorithm reaches performance very close to SAC Haarnoja et al. (2018). As pointed out in 4.3, we also note that a typical pattern with ADLoss is an early peak towards higher return (indicative of accelerated policy learning).

**Adapted PbRL Baselines:** While we have discussed that action distance loss provides the reward model about valuable information like state space structure, reachability, environment dynamics, part of these could also be given by other auxiliary tasks like forward dynamics prediction Nguyen et al. (2021); Zhang et al. (2018) which were proposed in the context of RL for improved policy learning. We adapt the task of forward dynamics prediction by updating the reward model embedding to be predictive of the next state (for the given state-action pair). We refer to this as "*Rdynamics*". Bisimulation metrics based representation learning Kemertas & Aumentado-Armstrong (2021) has also been effective for policy learning. Since bisimulation metrics, by definition, require a reward function we adapt it to use predicted reward values for PbRL as described in F.3. *Rdynamics* and *BISIM* baselines are essentially dynamics dependent baselines. On the other hand we consider a dy-

namics independent baseline - *L2EmbeddingLoss*. As euclidean distances can also be preserved in the embedding space by the stress loss in mMDS (Equation 4), we create a baseline "*L2EmbeddingLoss*" that uses L2 distance between the original state representations as the target in mMDS. Fig. 3 show that Rdynamics, BISIM, and L2EmbeddingLoss, although are a slight improvement over PEBBLE, are clearly weaker approaches than action distance based auxiliary task. Since it is known that learning world models can be a challenging Ha & Schmidhuber (2018) task in itself, then, even though Rdynamics captures information about the environment dynamics, incorporating it into PbRL is nontrivial. While we adapt Bisimulation metrics into PbRL, it essentially uses the predicted reward function (non-stationary and possibly noisy estimate) to improve its own embedding possibly leading to marginal improvements over baseline PEBBLE. Finally, we note that L2EmbeddingLoss provides no additional useful inductive bias (for e.g., state space structure in our case, SSL over trajectories in SURF, etc.) to provide any basis for improvement, and performs the worst.

## 5.2 Additional Experiments

**Human Study:** In our human study, we tested ADLoss vs PEBBLE baseline on three tasks: "window-open" and "door-open" from Meta-World, and "quadruped" from DMControl. In the "quadruped" task, the objective was to raise the front-right leg. Appendix (I) provides the study details, interface information, and complete results. While HiL is able to successfully train both the PEBBLE and ADLoss agents and express their preferences, we observed that the ADLoss-based PbRL agent consistently required fewer feedbacks on average (window-open: 336/593, door-open: 2033/2693, lift-front-right-leg: 126/173).

**Ablations and other experiments :** We conducted additional ablations on Meta-World-Button-Press and Sweep-Into. **Effect of Query length:** From Fig. 4 (left), we find that minor change to the length of the query segment dose not greatly impact the agent's performance. This is important as although the given query sizes are inconsequential with respect to compute, these size differences can have a huge cognitive impact on HiL. Fig. 4 (center) tests ADLoss performance at **different $\lambda_{ad}$ values** in the ADLoss reward learning objective (Eq. 8). Similar to Park et al. (2022), we find that tuning this hyperparameter has the most impact on the performance. Next, we test our ADLoss in conjunction with a SSL technique (proposed in Appendix G) to study **how compatible ADLoss is to other PbRL approaches**. As discussed in Section 4.3, we find that the SSL approach (TLoss), when used in conjunction with ADLoss benefits from accelerated policy learning. Finally, for completeness, we test the **impact of incorrect labels with mistake-oracle** Lee et al. (2021b) on ADLoss and find that it's robustness to such noise is no worse than baseline PEBBLE.( See Appendix H).

## 6 Discussion

In this work, we present a Preference-based Reinforcement Learning algorithm that can leverage learning signals form the joint policy being learned to provide valuable information like reachability between states and environment dynamics. We extract this learning signal via action distances, which is the expected number of actions taken under a policy to go from one state to another, and incorporate it into the reward learning objective by proposing an auxiliary objective for the reward model to be predictive of these action distances. This auxiliary objective ensures that the Euclidean distance between the embeddings (in the reward model) between two (state,action) tuples reflects their action distance. In addition to aiding the reward learning process, we show that action distance heuristics are pessimistic which provides a well founded intuition to explain accelerated the policy learning. Our experiments on six Meta-World robotic manipulation tasks shows the effectiveness of our approach over several PbRL baselines.

While action distances are proposed and utilized in the context of PbRL, interesting future work can study the use of action distances beyond reward learning. While PbRL suffers from high sample complexity and extremely weak feedback signal, the benefits of action distances may be pronounced in this setup and further investigation on the role of action distances beyond PbRL can be fruitful. While we showcase the benefits of action distance based auxiliary task for reward learning, another key limitation is on the potential scalability in highly stochastic environments where action distance estimates may become extremely noisy and hard to learn.

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

APPENDIX TO PAPER :
EXPLOITING ACTION DISTANCES FOR REWARD LEARNING FROM HUMAN
PREFERENCES

## A   ETHICS STATEMENT

This paper focuses on improving Preference-based Reinforcement Learning (PbRL) agents and identifies valuable learning signals (hinting at goal directed information based on binary human preference feedback). We introduce an action distance measure derived from the policy and utilize it as an auxiliary prediction task for reward learning. The evaluation of the proposed approach includes simulated human oracles and human-in-the-loop experiments across different tasks.

The ethical implications of this research lie in its potential impact on privacy, adherence to the intended preference, consent, transparency to name a few. While this research attempts to reduce the amount of feedback required from human in the loop and effectivelly reduces the overall cognitive load on the human (including the fatigue caused due to being actively present in the training loop), taking into account other ethical considerations like consent, privacy and transparency are still active research areas in PbRL community.

In the experimental design with the human in the loop, we ensure that human participants are fully informed about the study's objectives and procedures. We obtain an informed consent, and participants have the right to withdraw from the study at any time. We anonymize the collected data before analyzing them. However, the PbRL community is still looking for ways to restrict the leak of private and confidential information from the collected feedbacks. Real world application of PbRL must take these into account as unknown potential biases may arise.

Transparency and interpretability of the learned policies and models should be promoted to facilitate understanding and trust. Past work in explainable AI has attempted to mitigate these issues but preference learning specific solutions should also be proposed. One potential direction for PbRL systems is project the learned reward function in human understandable visualization / language which can serve as a means to check whether the learned preferences are in fact what the human had intended in the first place. While binary preference feedbacks have many flaws, they do restrict the amount of information conveyed by the human thereby limiting the leak of private / confidential information.

Finally, interesting future work may include design of specialized oracles that can simulate users that have specific biases and confidential strategic information baked into their preferences and verify whether in the process of learning human preferences such information is also captured by the agent. If so, prior work in the field of ethics and AI can serve as useful tools to mitigate or control the flow of such information.

## B   PROOFS

### B.1   PROOF FOR PROPOSITION 4.6

**Statement:** *For a stochastic policy $\pi$ that induces a stationary distribution, under a balanced sampling from the dataset obtained by from Algorithm 2, a perfect function approximator can estimate the action distance between the states $s_i$ and $s_j$.*

**Proof Sketch:** Algorithm 2, in the limit, can generate infinitely many samples for action distance targets from a state $i$ to $j$, and $j$ to $i$. This is easy to realize as the dataset is constructed by uniformly picking $s_i, s_j$ pairs from trajectories sampled via the policy $\pi$. As the action distance estimate is the average of the distance from $i$ to $j$ and back. It is possible that the number of data points sampled from $i \rightarrow j$ are considerably more than $j \rightarrow i$ in which case even a perfect approximator under MLE (Maximum Likelihood Estimation) assumption would predict the mean of the observed samples as the action distance. The mean of the observed samples is guaranteed to be equal to the commute distance or the action distance if the observed samples along each direction $i \rightarrow j$ and $j \rightarrow i$ are balanced Venkattaramanujam et al. (2019).

## C    PBRL ALGORITHM

We present our PbRL algorithm, as shown in Lee et al. (2021a), which uses the PEBBLE as a backbone. The integration of action distance loss into the PbRL algorithm requires no change to the model architectures or the learning paradigm and can be easily done so by updating the lines in red in Algorithm 1. SURF updates the same parts of the pseudocode as ours where the SSL (semi-supervised learning) approach in SURF integrates in to the $\mathcal{L}^{Reward}$ and the SSL step populates the feedback buffer with pseudo-labels.

---

**Algorithm 1** Integrating ADLoss into PEBBLE

---

**Input:** feedback frequency $K$, # queries per feedback session $M$
Initialize parameters of $Q_\theta$ and $\hat{r}_\psi$
Initialize a dataset of preferences $D_h \leftarrow \varnothing$
// EXPLORATION PHASE
$D_\tau, \pi_\phi \leftarrow$ EXPLORE() in Lee et al. (2021a)
POLICY LEARNING PHASE
**for** each iteration **do**
  // REWARD LEARNING PHASE
  **if** iteration % $K$ == 0 **then**
    **for** m in 1...$M$ **do**
      $(\sigma^0, \sigma^1) \sim$ SAMPLE() in Lee et al. (2021a)
      Query instructor for $y$
    **end for**
    **for** each gradient step **do**
      Sample mini-batch $\{(\sigma^0, \sigma^1, y)_j\}_{j=1}^{D_h} \sim D_h$
      Perform Semi-Supervised Learning as in Algorithm 2
      Optimize $\mathcal{L}^{Reward}$ w.r.t. $\psi$
    **end for**
    Relabel entire replay buffer $D_\tau$ using $\hat{r}_\psi$
  **end if**
  REINFORCEMENT LEARNING PHASE
  **for** each time-step $t$ **do**
    Collect $s_{t+1}$ by taking $a_t \sim \pi_\phi(a_t|s_t)$
    Store transitions $D_\tau \leftarrow D_\tau\{(s_t, a_t, s_{t+1}, \hat{r}_\psi(s_t))\}$
  **end for**
  **for** each gradient step **do**
    Sample random mini-batch $\{\tau_j\}_{j=1}^{D_\tau} \sim D_\tau$
    Optimize $L_{critic}^{SAC}$ and $L_{actor}^{SAC}$ w.r.t. $\theta$ and $\phi$, respectively, as in Lee et al. (2021a)
  **end for**
**end for**

---

### C.1    PEBBLE ALGORITHM

PEBBLE is a PbRL algorithm that comprises of two key elements: pre-training and relabeling experience buffer. To gather a wide range of experiences, PEBBLE starts by using intrinsic motivation Chentanez et al. (2004); Barto (2013); Abel et al. (2021); Schmidhuber (2010) to pre-train the policy, which optimizes the policy to increase the state entropy in order to explore the environment effectively. Afterwards, PEBBLE uses the SAC algorithm, a state-of-the-art off-policy RL algorithm, to further train the policy. To ensure stability in the learning process, PEBBLE relabels all experiences in the buffer when the reward model is updated.

### C.2    DATASET FOR ACTION DISTANCE LOSS UPDATE

Algorithm 2 describes the algorithm used to collect the dataset of state pairs and corresponding action distance targets.

---

**Algorithm 2** Dataset for Action Distance Loss Update

---

**Input:** $D_\tau$, Recent window $k$, Samples per trajectory $n$
$D_\tau^k = D_\tau[k \cdots N]$ {$N$ is size of $D_\tau$}
Initialize $D_{ad} \leftarrow \varnothing$
**for** $ix = 0$ **to** $N - k$ **do**
    **for** $iy = 0$ **to** $n$ **do**
        Uniformly choose $i, j$ s.t. $i < j \leq T$ {T} is length of trajectory $\tau$
        $x, y = ((s_i^\tau, a_i^\tau), (s_j^\tau, a_j^\tau)), |j - i|$
        $D_{ad} = D_{ad} \cup (x, y)$
    **end for**
**end for**
**return** $D_{ad}$

---

## D  EXPERIMENT DETAILS

Unless stated otherwise, we have attempted to keep all the hyperparameters & experiments settings as close to that proposed in prior works Park et al. (2022); Lee et al. (2021a).

### D.1  STATE SPACE AND ACTION SPACE

We use the available Meta-world package Yu et al. (2020) to instantiate our environments. We use the default state space representation given by the package, that contains information about the Cartesian position of the end-effector, positions of objects of concern etc. The details about the action space and the observation space are given in Yu et al. (2020).

### D.2  REWARD ARCHITECTURE AND EMBEDDING SPACE

Following the implementation of Lee et al. (2021a); Park et al. (2022) we implement reward model via a neural network and bound the final output using a $\tanh$ activation function : [-1, 1]. For all the Meta-world experiments the reward model has three hidden layers with 256 neurons each followed by the prediction layer (with one neuron). The embedding space used for minimizing the metric Multidimensional Scaling stress or the derived ADLoss Mean squared error is the penultimate layer of the network. We use the ADAM optimizer for training the SAC actor-critic as well as the reward model. The hyperparameters used for baseline PEBBLE, SURF and ADLoss (our action distance based auxiliary task loss) are given in tables 3, 4, and 5.

The input to the reward model is (state, action) tuple. As used previously Park et al. (2022); Lee et al. (2021b), we stack the state and action vectors and treat them as a single input for the reward model.

### D.3  ORACLE

While B-Pref Lee et al. (2021b) explores various types of scripted humans in the loop like myopic, noisy etc, since our primary objective in this work is to evaluate the effectiveness of Action Distance measure for PbRL, we assume an oracle scripted human who uses the underlying reward to correctly provide the binary feedback. The feedback is given as follows :

$$y(\tau_0, \tau_1) = \begin{cases} 0, & \text{if} \quad \sum_i \tilde{R}_h(\tau_0) > \sum_i \tilde{R}_h(\tau_1) \\ 1, & \text{if} \quad \sum_i \tilde{R}_h(\tau_0) < \sum_i \tilde{R}_h(\tau_1) \end{cases} \tag{9}$$

where, $\tilde{R}_h$ is the environment reward being used as the human preference reward function and $\tau_0, \tau_1$ are the queried trajectory pairs. Note that we work under the setting that the preference feedback is binary and therefore if the trajectory returns are equal we uniformly pick a preferred trajectory. This does not pose any problems with our chosen benchmark domains as the underlying reward is dense and shaped Devlin & Kudenko (2012); Ng et al. (1999).

### D.4 SAMPLING SCHEMES

We refer the readers to Lee et al. (2021a); Christiano et al. (2017) for the various sampling schemes that have been proposed in prior work. In this work, for all the experiments we use the disagreement based sampling for selecting pair of trajectories to query to the HiL.

### D.5 IMPLEMENTATION, CODE AND COMPUTE

We use the publicly available implementation of B-Pref Lee et al. (2021b) for the implementation of PEBBLE and SAC. We implement the remaining baselines, SURF, Rdynamics, L2EmbeddingLoss (code in supplementary). All the experiments were run on an Intel(R) Xeon(R) Gold 6258R CPU @ 2.70GHz, with Quadro RTX 8000 GPU.

### D.6 EVALUATION DOMAINS

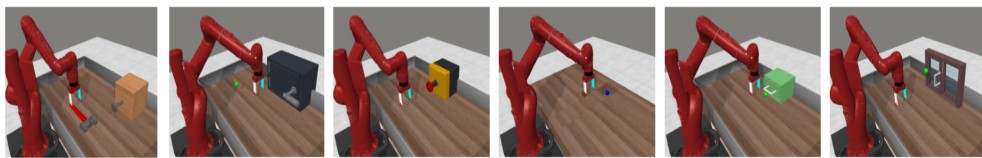

(a) Hammer     (b) Door Open   (c) Button Press   (d) Sweep Into   (e) Drawer Open (f) Window Open

Figure 5: Rendered images of the Meta-world Yu et al. (2020) evaluation domains.

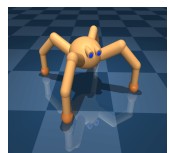

(a) Quadruped

Figure 6: Rendered image of DMControl Suite Tunyasuvunakool et al. (2020) Quadruped for Human in the loop study

We present here in Table 1, the description of the Meta-World domains that we have used to show our empirical evaluations, along with the respective environment rewards in Table 2 as have been specified in Yu et al. (2020).

Table 1: Meta-World domain descriptions, as in Yu et al. (2020).

| Task | Description |
|------|-------------|
| Hammer | Hammer a screw on the wall. Randomize the hammer and the screw positions. |
| Door Open | Open a door with a revolving joint. Randomize door positions. |
| Button Press | Press a button. Randomize button positions. |
| Sweep Into | Sweep a puck into a hole. Randomize puck positions. |
| Drawer Open | Open a drawer. Randomize drawer positions. |
| Window Open | Push and open a window. Randomize window positions. |
| Unlock Door | Unlock the door by rotating the lock counter-clockwise. Randomize door positions. |
| Plate Slide | Slide a plate into a cabinet. Randomize the plate and cabinet positions. |

Table 2: Meta-World domain rewards, as in Yu et al. (2020).

| Task | Description |
|------|-------------|
| Hammer | $- \parallel h - o \parallel_2 + I_{\parallel h - o \parallel_2 < 0.05} \cdot 100 \cdot \min\{o_z, z_{target}\}$ |
| | $+ I_{\mid o_z - z_{target} \mid < 0.05} \cdot 1000 \cdot \exp\{\parallel h - g \parallel_2^2 / 0.01\}$ |
| Door Open | $- \parallel h - o \parallel_2 + I_{\parallel h - o \parallel_2 < 0.05} \cdot 1000 \cdot \exp\{\parallel h - g \parallel_2^2 / 0.01\}$ |
| Button Press | $- \parallel h - o \parallel_2 + I_{\parallel h - o \parallel_2 < 0.05} \cdot 1000 \cdot \exp\{\parallel h - g \parallel_2^2 / 0.01\}$ |
| Sweep Into | $- \parallel h - o \parallel_2 + I_{\parallel h - o \parallel_2 < 0.05} \cdot 1000 \cdot \exp\{\parallel h - g \parallel_2^2 / 0.01\}$ |
| Drawer Open | $- \parallel h - o \parallel_2 + I_{\parallel h - o \parallel_2 < 0.05} \cdot 1000 \cdot \exp\{\parallel h - g \parallel_2^2 / 0.01\}$ |
| Window Open | $- \parallel h - o \parallel_2 + I_{\parallel h - o \parallel_2 < 0.05} \cdot 1000 \cdot \exp\{\parallel h - g \parallel_2^2 / 0.01\}$ |
| Unlock Door | $- \parallel h - o \parallel_2 + I_{\parallel h - o \parallel_2 < 0.05} \cdot 1000 \cdot \exp\{\parallel h - g \parallel_2^2 / 0.01\}$ |
| Plate Slide | $- \parallel h - o \parallel_2 + I_{\parallel h - o \parallel_2 < 0.05} \cdot 1000 \cdot \exp\{\parallel h - g \parallel_2^2 / 0.01\}$ |

# E HYPERPARAMETERS

Table 3: Hyperparameters of backbone PEBBLE in our experiments.

| Hyperparameter | Value | Hyperparameter | Value |
|---|---|---|---|
| Initial temperature | 0.1 | Hidden units per each layer | 1024(DMControl), 256(Meta-world) |
| Length of segment | 50 | # of layers | 2(DMControl), 3(Meta-world) |
| Learning rate | 0.0003 (Meta-world) | Batch Size | 1024(DMControl), 512(Meta-world) |
| | 0.0005 (Walker) | Optimizer | Adam Kingma & Ba (2014) |
| | 0.0001 (Quadruped) | | |
| Critic target update freq | 2 | Critic EMA $\tau$ | 0.005 |
| $\beta_1, \beta_2$ | (0.9, 0.999) | Discount $\gamma$ | 0.99 |
| Frequency of feedback | 5000 (Meta-world) | Maximum budget / | 1000/100, 100/10(DMControl) |
| | 20000 (Walker) | # of queries per session | 10000/50, 4000/20(Meta-world) |
| | 30000 (Quadruped) | | 2000/25, 400/10 (Meta-world) |
| # of ensemble models $N_{en}$ | 3 | # of pre-training steps | 10000 |

Table 4: Hyperparameters of SURF.

| Hyperparameter | Value |
|---|---|
| Unlabeled batch ratio $\mu$ | 4 |
| Threshold $\tau$ | 0.95 |
| Loss weight $\lambda$ | 1 |

Table 5: Additional hyperparameters used in our experiments.

| Hyperparameter | Value |
|---|---|
| Recent trajectories $k$ | 5 |
| ADLoss weight | 20 |

# F ADDITIONAL EXPERIMENT DETAILS

## F.1 SETUP FOR RDYNAMICS BASELINE

The Rdynamics baseline requires modification to the reward model. While imposing assumptions that one can easily modify the model architecture should be avoided, predicting forward state dynamics has been used in past RL literature and has been shown to be useful for policy learning. Just as we use the penultimate reward model layer as the embedding space, we attach another layer that predicts the next state and add a forward state prediction loss. Fig. 7 illustrates the reward architecture used for Rdynamics baseline and below is the additional loss used for forward state prediction, where $y$ is the predicted next state. A linear combination of $\mathcal{L}_{forwarddynamics}$ and $\mathcal{L}_{CE}$ (similar to ours, in equation 8) gives the reward learning objective for Rdynamics.

$$\mathcal{L}_{forwarddynamics} = MSE(y, s_{t+1}) \tag{10}$$

## F.2 SETUP FOR L2EMBEDDINGLOSS BASELINE

L2EmbeddingLoss uses the same $\mathcal{L}^{Reward}$ objective as in Equation 8, except that in Equation 7 the targets $d_y$ are set to be, $||s_j - s_i||^2$, i.e., the L2 norm of the difference between the two states.

## F.3 SETUP FOR BISIM BASELINE

We use the following equation for adapting bisimulation metrics into the PbRL loop :

$$\mathcal{J}_{bisim}(\psi) = (||z_i - z_j||_1 - |r_i - r_j| - \gamma||z_i^{'(z_i, a_i)} - z_j^{'(z_j, a_j)}||)^2, \tag{11}$$

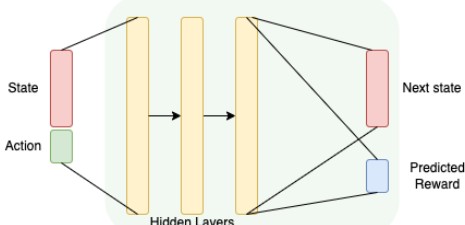

Figure 7: Overview of the reward model architecture for Rdynamics.

adapted from Eq. 4 in Kemertas & Aumentado-Armstrong (2021). For a state pair $s_i, s_j$, we use the learned reward model to obtain the predicted rewards $r_i, r_j$ in the equation above. Similar to the other baselines RDynamics and L2EmbeddingLoss we use the penultimate layer as the embedding layer to obtain $z_i, z_j$. Similar to RDynamics we add a head from the penultimate layer that predicts the next state embedding to obtain $z'_i, z'_j$. Finally, we reuse the trajectory buffer (as in ADLoss) to obtain the states on which we compute the bisimulation-target distance to optimize the above loss.

# G   TRIPLET LOSS (TLOSS) BASED SEMI-SUPERVISED LEARNING OBJECTIVE

While we have shown the benefits of ADLoss in section 5, we wanted to test how would ADLoss perform in conjunction with another SSL approach.

In pursuit of proposing a comprehensive, yet complementary solution for PbRL, we present a novel semi-supervised learning objective (SSL) that utilizes pseudo-labelling for unlabelled trajectories followed by a triplet loss minimization that can be used in conjunction with the proposed Action Distance Auxiliary task. The proposed combination explores the compatibility of action distances based auxiliary task with other add-on PbRL techniques.

In our limited testing we have found that the combination of ADLoss and the following triplet loss SSL outperforms the PEBBLE / SURF baselines and is complementary across several domains

This SSL objective is operationalized by a triplet loss that requires a specified anchor data point, and positive and negative samples. It then ensures that there exists at least some margin gap between the +sample-anchor distance and -ve-sample-anchor distance. We first set the sampled trajectory from the unlabelled dataset as our anchor. The mini-batch of trajectory pairs sampled from the human feedback replay buffer will serve as positive or negative samples. To do so, we make an Absolute Preference over Mini-batch assumption as,

**Assumption G.1.** In a mini-batch of $k$ samples from human feedback buffer as triplets of $(\tau_0, \tau_1, y)_{i=1}^{i=k}$, the preference label can be treated as an absolute preference about the trajectory, i.e. if $y = 1$ then trajectory is a preferred trajectory (not relative to other trajectories).

Assumption G.1 enables the creation of a bipartite set of trajectories, with set $g$ containing all preferred trajectories and set $b$ containing all dis-preferred trajectories. If the queries are made from far enough regions of the state space, the assumption holds well, but if the batch size is too large, it is possible to encounter situations where the assumption does not hold. Because of this, we limit the batch size to be in the set $\{8, 16, 32\}$.

We perform a pseudo-labelling step where we identify whether the sampled trajectory $\tau$ is closer to the trajectories in set $\mathbf{g}$ or $\mathbf{b}$ where the label is:

$$\mathbf{y} = \begin{cases} \mathbf{g} & ; d(\mathcal{R}(\tau), \underset{g \sim \mathbf{g}}{\mathbb{E}}[\mathcal{R}(\tau_g)]) < d(\mathcal{R}(\tau), \underset{b \sim \mathbf{b}}{\mathbb{E}}[\mathcal{R}(\tau_b)]) \\ \mathbf{b} & ; \text{otherwise} \end{cases} \tag{12}$$

where $d$ is the $L_2$ distance function. We use the set $\mathbf{y}$ as the positive set of data points and $\mathbf{1\text{-}y}$ as the negative set for our triplet loss,

$$\mathcal{L}^t(\tau) = max(0, ||\mathcal{R}(\tau) - \underset{y \sim \mathbf{y}}{\mathbb{E}}[\mathcal{R}(\tau_y)])||^2$$
$$-||\mathcal{R}(\tau) - \underset{\tilde{y} \sim \mathbf{1\text{-}y}}{\mathbb{E}}[\mathcal{R}(\tau_{\tilde{y}})])||^2 + m) \tag{13}$$

where $m$ is the margin hyperparameter. We overload the notation for reward to reflect the rewards for the trajectory states as a vector, i.e., $\mathcal{R}(\tau) = [R(s_0) \quad R(s_1) \quad \cdots \quad R(s_{T-1}) \quad R(s_T))]^T$.

Figure 4 Sweep-Into shows results of using TLoss, ADLoss and Cross Entropy together and ablates against each component. We find that while the performance of the agent (ADLoss + TLoss) is quite close to the best performer, the addition of action distance based loss to the triplet loss does accelerate policy learning - a feature which was theoretically motivated for action distances in Section 4.3.

# H    MISTAKE ORACLE

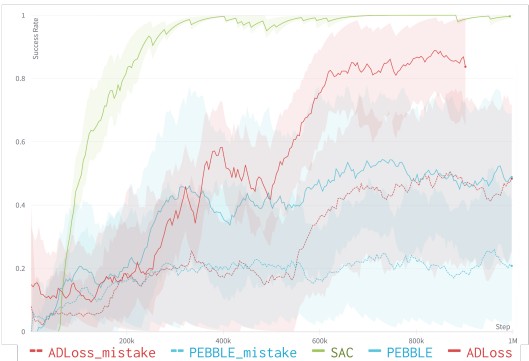

Figure 8: Using a simulated mistake oracle on Sweep Into domain, and using SAC once again as the upper bound on the success rate, we note that the success rate drops for ADLoss (with mistake oracle). However, it in fact performs equally well as the original PEBBLE algorithm Lee et al. (2021a).

## H.1    SETUP

Existing works Lee et al. (2021a;b) often suffer from poor performance when teachers can provide the wrong labels. It is safe to assume that humans can make mistakes when prompted to give their feedback, and hence, we test our algorithm for robustness using a simulated mistake oracle. A mistake oracle is computed by flipping a trajectory preference by probability $\epsilon$, similar to the definition in Lee et al. (2021b). Furthermore, the mistake oracle can be instantiated as ($\beta \to \infty$, $\gamma \to 1$, $\epsilon \to 0.1$, $\delta_{skip} \to 0$, $\delta_{equal} \to 0$), where $\beta$ is the rationality constant, $\gamma$ is the discount factor, $\delta_{skip}$ is the skip threshold, and $\delta_{equal}$ is the equal threshold, as in Lee et al. (2021b).

## H.2    INTUITION

As true for several supervised data-driven approaches, noisy supervision can indeed impact the agent performance (as confirmed in Lee et al. (2021b)). Semi-Supervised learning is one of the typical methods to mitigate negative effects of noisy labels by learning a more robust representation of data. While we do not directly generate labels in semi-supervised learning or specifically make claims regarding robustness to noisy labels, the action distance computed does indicate distance between states (in a semi-supervised way) notion and enjoys a much larger dataset for learning the auxiliary task. The RL backbone used in our work is SAC which learns a stochastic policy. This implies that the action distance targets being used for learning the auxiliary task already have some variance (and therefore is already robust to some extent). While noisy labels can further affect the individual distance targets, we believe that so long the expectation does not change too much, the action distance based auxiliary task should be helpful as it still is a useful heuristic. This engenders belief that ADLoss-based solution should be more robust to noise than baselines, if not similar.

## H.3    RESULTS

We test the mistake oracle Lee et al. (2021b) with ADLoss and PEBBLE. Figure 8 shows that mistake oracle can have a severe impact on the agent's performance. While we do test ADLoss with real human in the loop (section I), for completeness, we also test it on simulated mistake oracle. As expected, both ADLoss and PEBBLE are negatively impacted, however the impact on ADLoss is no worse than PEBBLE. In fact, the performance of ADLoss with the mistake oracle is at par with baseline PEBBLE with perfect oracle.

## I HUMAN STUDY

### I.1 USER STUDY SETUP

We conduct a user study with 3 participants, each participant interacting with 3 environments, i.e., Quadruped: lift front right leg, Door Open and Window Open, and 2 algorithms, i.e., PEBBLE Lee et al. (2021a) and ADLoss. Hence, we have a total of 18 experiment runs with human in the loop where collect the total number of feedbacks required in each of these runs. The results have been shown in Table 6.

The participants provide their binary preferences using the interface shown in I.3 for a fixed number of queries per session. At the end of each session they are shown trajectory pairs sampled from a recent episode to indicate the current performance of the agent. Users are expected to use this last web page to conclude their participation, should they feel that the training is complete. Participants are also free to end the training if they are fatigued etc.

### I.2 RESULTS

As shown in Table 6, we see that for each (human, environment) experiment run, ADLoss requires significantly lesser number of feedbacks. On an average, ADLoss takes $\approx 257$ less feedbacks on Window Open, $\approx 660$ less feedbacks on Door Open, and $\approx 47$ less feedbacks on the Quadruped task.

Table 6: Results from our user study.

| Environment | PEBBLELee et al. (2021a) | | | | | ADLoss | | | | |
|---|---|---|---|---|---|---|---|---|---|---|
| | Human 1 | Human 2 | Human 3 | Avg. | Std. | Human 1 | Human 2 | Human 3 | Avg. | Std. |
| Window Open | 620 | 580 | 580 | 593.3 | 23.1 | 340 | 360 | 310 | 336.6 | 25.2 |
| Door Open | 2640 | 3000 | 2440 | 2693.3 | 283.8 | 2020 | 2120 | 1960 | 2033.3 | 80.8 |
| Lift Front Right Leg | 180 | 180 | 160 | 173.3 | 11.5 | 120 | 120 | 140 | 126.6 | 11.5 |

### I.3 USER STUDY INTERFACE

For conducting the human in the loop experiments, we first provide a consent form, as shown in Figure 9a where we mention about the IRB Approval used for the study, and provide a task description to each user. We create a user interface, as shown in Figure 9b, to record the interaction of the human in the loop with our framework during the course of this study. Note, that the information regarding the IRB Approval and author details have been redacted for ensuring anonymity.

In the user interface shown in Figure 9b, the human in the loop is shown two trajectories that are sampled for querying, and they can use keyboard shortcuts for choosing either the left or the right trajectory. The interaction ends when the human closes the browser. In the final query of each session, the human is also asked if the preferred trajectory matches their expectation. Answers to this question are not used by the algorithm and are collected for future work.

### I.4 HUMAN SUBJECT DETAILS

The users have a median age of 25 and mean age of 25. All the participants are university graduate students and have previously taken AI related coursework in their study, however only one of the three participants (Human 3) was familiar with the domains used in the experiments.

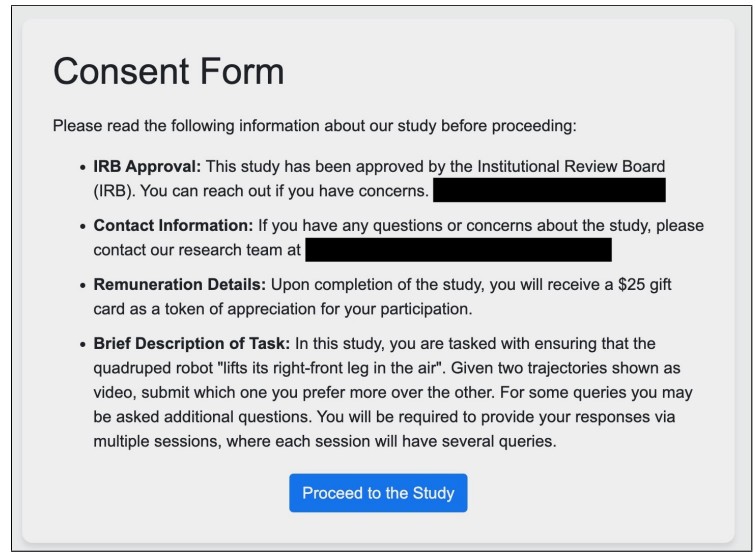

(a) User Study Consent Form

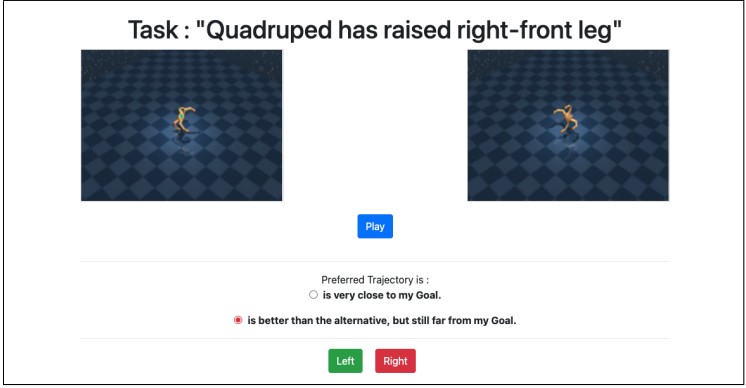

(b) User Interface for task: raised right-front leg.

Figure 9: Screenshots from our user study consent form (author information redacted for anonymity) and user interface (UI) to conduct the study on the Quadruped domain.

