# OpenReview forum: "Exploiting Action Distances for Reward Learning from Human Preferences"
_ICLR.cc/2024/Conference — Submitted to ICLR 2024_

### Official Review · Reviewer_Yyxw · 2023-10-26

**Soundness:** 2 fair
**Presentation:** 2 fair
**Contribution:** 2 fair
**Rating:** 3
**Confidence:** 5

**Summary:**

This paper proposes to augment the reward function learning in Preference-based RL (PbRL) with an auxiliary objective of predicting the number of steps it takes to reach other states.

**Strengths:**

- Adding auxiliary losses to the reward function learning in PbRL is an important idea that can lead to sample and feedback efficiency improvements.
- The paper is written well in a language that is easy to understand and follow.
- The user study is appreciated and shows that indeed the use of ADLoss makes the learning require fewer number of feedbacks.

**Weaknesses:**

## The key premise — Using policy for reward model may not always work
- In RL, the reward acts as the only learning signal and in PbRL, preference determines the reward function. While improving the reward model training by the use of environment dynamics makes sense, reachability via action distance is an artifact of both the original state distances and also the currently trained policy. This could induce a non-stationarity in the learning process because of the cyclic dependency of the reward on the policy and the policy on the reward. So, I have concerns about the robustness of the heuristic of action distance proposed in this work.
- What is the support for the hypothesis that a policy trained over learned rewards tells more about the human preference. This is quite counter-intuitive because the reward is the only signal to train the policy (it acts like a bottleneck interface between the policy and the human preference), so how can the policy have more information to return to the reward function, about the human preference?
- Can the authors justify why the ADLoss would never hurt reward learning in any environment?

## Experiment Concerns
- Inconsistent experiment settings from PEBBLE paper: Is there a reason why the number of feedbacks chosen in this work are different from the base settings provided in the PEBBLE paper? Also, why are the set of 6 Metaworld tasks chosen different?
- Why does PEBBLE converge suboptimally? Are the parameters for PEBBLE well-tuned? What was the scheme of hyperparameter optimization? Can the authors report results on the exact same environment setups used in PEBBLE, so the baseline results can be verified and trusted that a fair amount of hyperparameter tuning has been performed? Then using the same hyperparameters, provide the results of ADLoss + PEBBLE.
- Since the proposed solution is a simple heuristic, it requires more thorough experimental validation than the the current results. It is hard to convincingly believe the experimental results being generally better than the baselines. The evaluation should be done on more Metaworld tasks and at least on the settings that PEBBLE reported if not more (because the effect of auxiliary loss could be negative — so it should be known when it works and when it does not work).
- Can the authors demonstrate some experiments with noisy dynamics, to show the robustness of the action distance Loss in expectation?
- No evaluation curves provided from the human study. At the stopping criterion chosen by humans, is it verified that the performance of PEBBLE and ADLoss are equivalent?

## Motivation of the approach and Baselines
- It is not clear why this particular form of ADLoss and this particular way of using it as an auxiliary objective is the best solution to the question of "how to make reward model learning more feedback efficient by incorporating the policy". As of now, it seems like a ad-hoc proposal that just happens to work, but it requires more thorough intuition and experimentation to show that this is the best way to answer this question. For instance,
	+ The authors mention that using a scaled linear combination is exteremely non-trivial and challenging. Why is that so? In fact, Equation (8) also has scaling hyperparameters between the loss terms. So, using an additional reward term is a valid baseline, and an important one to justify the need for auxiliary loss.
- The motivation from Cheng et al. (2021) is not applicable because they modify the reward function and the discount factor with pessimistic heuristics, and the action distance being a pessimistic measure of the reachability of two states, and then using it as an auxiliary task, has nothing to do with their results, even in the goal-conditioned case.
- Ideally, SURF + ADLoss v/s SURF should be shown, to show ADLoss is complementary.
- Why would auxiliary loss over the state representation help? It is not like the states are image-based, so representation learning doesn't seem like the most crucial challenge here. It has to be providing some task-specific information, in which case, there must be more direct ways of utilizing this action distance idea.

## How about alternative auxiliary losses?
- Motivation for ADLoss is lacking: there can be many more baselines compared against for auxiliary losses, including Inverse Dynamics Prediction (or Curiosity Loss) or Contrastive Loss or Self-Supervised Domain-specific losses or reconstruction.

**Questions:**

- How could we disentangle the effect of the current policy's training progress on the reward function? Because, this way, we might lead to suboptimal policies if we affect the learned reward model using inaccurate auxiliary signals. Why not standard auxiliary losses that would inform the state representations about the environment - dynamics and inverse dynamics losses? They would not have reachability information and would only have the information about the environment.
- Is the action distance just a reachability measure? Were other reachability metrics tried?
- What are other ways to get learning signal from the policy or otherwise? Would self-supervised learning as an auxiliary objective also help equivalently or complementarily?
- Why should the selection of action distance targets be from the most recent trajectories? Wouldn't this make the reward learning procedure non-stationary?

---

### Official Review · Reviewer_kRFF · 2023-10-27

**Soundness:** 2 fair
**Presentation:** 3 good
**Contribution:** 1 poor
**Rating:** 3
**Confidence:** 4

**Summary:**

This paper proposes an auxiliary objective for training the reward model in preference-based reinforcement learning (PbRL) methods. The idea is quite simple. Instead of using only the cross-entropy loss to train the reward model as in most PbRL methods, it uses an additional action distance based loss function to train the reward model. This action distance based loss function is used to constrain the embedding space in the reward model to preserve action distances, i.e., to ensure the Euclidean distance between the embedding of two states s_i and s_j reflects the action distance (i.e., the expected number of actions taken to reach s_j from s_i) between these two states. It is claimed that this can help capture information including the reachability of states and environment dynamics and in turn help improve policy learning.

**Strengths:**

- The paper is well-written and easy to follow overall.
- The proposed method is well-motivated, simple and general (in theory).

**Weaknesses:**

- While I appreciate that the proposed method is simple and general ("can be used with any existing PbRL technique" in theory), I think it is crucial to demonstrate the effective of the proposed method when combined with different existing PbRL algorithms. Currently, the proposed reward learning objective is only combined with PEBBLE, which is not enough. It should be combined with other existing PbRL methods and evaluated as well.
- The experimental results are not that convincing/competitive in my opinion:
	- Based on Figure 2, I think the proposed method ADLoss does not really perform significantly better than SURF in most tasks. ADLoss and SURF perform similarly in Hammer and Button. Their variances overlap in Door Open and Window Open.
	- In the experiments, the oracle scripted human uses the underlying true reward to provide perfect binary feedback. To demonstrate the robustness of the proposed method, I think it is important to test some more realistic models of human teachers (simulated human with various irrationalities).
	- The human user study only consists of 3 human participants. It is hard to draw any conclusion that is statistically useful.
- Some minor suggestions:
	- I think the method section is unnecessarily long. It includes a long discussion about different methods that could be explored but probably wouldn't work well, which could be moved to the appendix.
	- Calling the proposed method (combination of the proposed reward learning objective with PEBBLE) as ADLoss is misleading since it sounds like only the action distance based loss function is used (while the original cross-entropy loss is actually used).
	- The way the references are cited throughout the paper can be improved.

**Questions:**

- In Figure 3, why the learning curve for ADLoss stops early (before reaching 1 million step) in Sweep Into, while all other methods reach 1 million step?
- Have you tried using other existing PbRL methods (other than PEBBLE) as the backbone algorithm to evaluate the effectiveness of using the proposed reward learning objective? It would be very helpful to see some additional experimental results regarding this.
- It would also be helpful to see some results of ADLoss using more realistic models of human teachers.

---

### Official Review · Reviewer_FW2T · 2023-10-31

**Soundness:** 2 fair
**Presentation:** 2 fair
**Contribution:** 2 fair
**Rating:** 3
**Confidence:** 4

**Summary:**

This paper proposes to improve the sample efficiency of Preference-based Reinforcement Learning (PbRL) algorithms by adding an auxiliary loss, ADLoss, to the reward learning objective. ADLoss encourages the reward function to embed information about the average number of actions needed to reach one state from another (which the paper properly defines as the action distance). ADLoss uses the mean-square error to ensure the distance between two embeddings is close to the respective action distance.

Through experiments with MetaWorld, and with a user-study, the paper shows improved performance, for instance in "Drawer Open" reaching the performance of SAC learned with a ground truth reward.

**Strengths:**

* The paper's intuition of using the policy-learning process to inform the reward-learning process is novel.
* The concept of action distance is well described and justified in the paper. As is the use of metric multi-dimensional scaling (mMDS) to justify the formulation of ADLoss is compelling.
* The paper ablations (including using ADLoss with a "mistake" labeller, an actual user study, verifying the effect of query length and ADLoss weighting) are thorough and useful to characterise ADLoss's behaviour.

**Weaknesses:**

* _W1_: I am very skeptical that the discussion about pessimistic heuristics actually explains the performance improvements of ADLoss. Per Section 4.3, there are several pre-requisites for ADLoss to be considered a pessimistic heuristic: the rewards must be $\le -1$,  the discount factor has to be $\gamma = 1$, the environment needs to be strong markovian. There is no attempt in the paper to show that either MetaWorld or DMC follow these strong pre-requisites. In my opinion, the intuitions given earlier in the paper about encoding dynamics, coupled with strong empirical evidence would be sufficient to prove the utility of ADLoss.
* _W2_: The empirical evidence needs to be strengthened. Current results show ADLoss improving upon SURF on only two environments: "Sweep Into" and "Drawer Open". Ideally, I would have liked to see experiments with DMC tasks (beyond the task in the user study), as well as experiments combining ADLoss with other sample efficiency PbRL algorithms (such as SURF and RUNE [1])



[1] X. Liang, K. Shu, K. Lee, and P. Abbeel. Reward uncertainty for exploration in preference-based reinforcement learning. ICLR 2022.

**Questions:**

* _Q1_: Could you include the final policy performance in the user study? Currently, only the amount of feedback is reported, and it is not clear whether PEBBLE, and PEBLE+ADLoss reach the same performance levels.
* _Q2_: In figures 1, 2 and 3, the curve for ADLoss for Sweep Into finishes abruptly. Could you update until 1M steps?
* _Q3_: I found the definition of classical MDS in equation (3) hard to parse. What are the weights $w_{ij}$? Also using $d_{ij}$ as both a function and as a variable is needlessly confusing.
* _Q4_: What is the advantage of metric MDS over classical MDS? Can you clarify what is $f(x)$ in ADLoss? It seems that the ADLoss objective in eq (7) maps cleanly into classical MDS.
* _Q5_: What is the effect in policy performance of using all combinations of states in a trajectory to estimate the action distance?

**Nitpicks and suggestions (will not affect rating)**

* When using information from the policy-learning process  for the reward-learning process, there is the risk that the reward overfits to the policy (or viceversa) and no learning happens. ADLoss cleverly avoids this by using only information about action distances. Do you agree? And if so, would you consider adding this point to the paper?
* Please use vector graphics in all your figures, they look blurry when zooming in.
* The plural form of feedback is "feedback" rather than "feedbacks", since it's an uncountable noun.
* At the beginning of section 2 I would suggest replacing "foremost" by "first" or "earliest".
* In section 3.1, $o$ is used for observation (which is typical of the POMDP formulation), but never refer to it again. Similarly $H$ is undefined.
* Please add references with sample usages of MDS in Section 3.2
* Consider moving Section 4.2 into the literature review (Section 2).
* In Figure 3, it may be better to include SURF rather than PEBBLE since SURF performs better in the tasks investigated.
* Consider adding performance tables for all experiments in the appendices. It's hard to estimate the mean and standard deviation from the plots.

---

### Official Review · Reviewer_tG87 · 2023-11-01

**Soundness:** 3 good
**Presentation:** 3 good
**Contribution:** 3 good
**Rating:** 6
**Confidence:** 3

**Summary:**

Authors introduce a preference based reinforcement learning algorithm that uses signals from the policy being learned to improve the reward model. This is done by learning the action distances using a policy function, which is then used to improve the preference RL by having the reward model capable of predicting this action distance. To this end, the reward learning also consists of an auxiliary action distance prediction task. The algorithm is then evaluated in 6 robot manipulation tasks. Authors further conduct a human-subject study, where 3 tasks were evaluated.

**Strengths:**

- Action distance presents a viable way to improve the sample efficiency of preference RL algorithms.
- Evaluations that include human studies and ablation studies provide wider understanding to the reader about the strength of the algorithm.

**Weaknesses:**

- The evaluations come from the same domain (meta-world), and in robotic manipulation. The evaluation can benefit greatly from introducing other domains (e.g. quadruped, walker).
- While the authors have conducted a human study, the number of participants (3, with 1 familiar with the domain) is low to evaluate any hypotheses.

**Questions:**

- As detailed above, adding more domains to the evaluation and increasing the participants in the human study can provide a better argument for the methods strengths.

---

### Meta-Review · Area_Chair_rUwE · 2023-12-06

**Metareview:**

Summary: This paper adds an auxiliary loss during preference-based learning called ADLoss: this encourages the reward function to embed information about the average number of actions needed to reach one state from another

Strengths: The paper is easy to follow, the idea is sound, the paper conducts experiments with users, and performs several ablations.

Weaknesses: Multiple reviewers mentioned that the empirical results do not show substantive improvements: e.g., “ADLoss does not really perform significantly better than SURF in most tasks.” Additionally, the human user study only consisted of 3 human participants, which is not a large enough sample from which to draw conclusions. Reviewers also raise the question of if other proxy losses could capture similar results (e.g., inverse dynamics prediction as is done in [1]).

[1] Daniel Brown, Russell Coleman, Ravi Srinivasan, and Scott Niekum. Safe imitation learning via fast bayesian reward inference from preferences. In International Conference on Machine Learning, pp. 1165–1177. PMLR, 2020.

**Justification For Why Not Higher Score:**

The proposed work does not perform significantly better than baselines and the human subject study is too small to draw conclusions from at this time (3 people).

**Justification For Why Not Lower Score:**

N/A

---

### Decision · Program_Chairs · 2024-01-16

Reject